# Online Markov Decoding: Lower Bounds and Near-Optimal Approximation Algorithms

**Vikas K. Garg**
MIT
vgarg@csail.mit.edu

**Tamar Pichkhadze**
MIT
tamarp@alum.mit.edu

## Abstract

We resolve the fundamental problem of online decoding with general $n^{th}$ order ergodic Markov chain models. Specifically, we provide deterministic and randomized algorithms whose performance is close to that of the optimal offline algorithm even when latency is small. Our algorithms admit efficient implementation via dynamic programs, and readily extend to (adversarial) non-stationary or time-varying settings. We also establish lower bounds for online methods under latency constraints in both deterministic and randomized settings, and show that no online algorithm can perform significantly better than our algorithms. To our knowledge, our work is the first to analyze general Markov chain decoding under hard constraints on latency. We provide strong empirical evidence to illustrate the potential impact of our work in applications such as gene sequencing.

## 1   Introduction

Markov models, in their various incarnations, have for long formed the backbone of diverse applications such as telecommunication [1], biological sequence analysis [2], protein structure prediction [3], language modeling [4], automatic speech recognition [5], financial modeling [6], gesture recognition [7], and traffic analysis [8, 9]. In a Markov chain model of order $n$, the conditional distribution of next state at any time $i$ depends only on the current state and the previous $n-1$ states, i.e.,

$$\mathbb{P}(y_i|y_1,\ldots,y_{i-1}) \;=\; \mathbb{P}(y_i|y_{i-n},\ldots,y_{i-1}) \;\forall i \;.$$

Often, the states are not directly accessible but need to be inferred or *decoded* from the observations, i.e., a sequence of tokens *emitted* by the states. For instance, in tagging applications [10], each state pertains to a part-of-speech tag (e.g. noun, adjective) and each word $w_i$ in an input sentence $\boldsymbol{w} = (w_1,\ldots,w_T)$ needs to be labeled with a probable tag $y_i$ that might have emitted the word. Thus, it is natural to endow each state with a distribution over the tokens it may emit. For example, $n^{th}$ order *hidden Markov models* ($n$-HMM) [11] and $(n+1)$-gram language models [4] assume the joint distribution $\mathbb{P}(\mathbf{y},\boldsymbol{w})$ of states $\mathbf{y} = (y_1,\ldots,y_T)$ and observations $\boldsymbol{w}$ factorizes as

$$\mathbb{P}(\mathbf{y},\boldsymbol{w}) \;=\; \prod_{i=1}^{T} \mathbb{P}(y_i|y_{i-n},\ldots,y_{i-1})\,\mathbb{P}(w_i|y_i) \;,$$

where $y_{-n+1},\ldots,y_0$ are dummy states, and the *transition* distributions $\mathbb{P}(y_i|y_{i-n},\ldots,y_{i-1})$ and the *emission* distributions $\mathbb{P}(w_i|y_i)$ are estimated from data. We call a Markov model *ergodic* if there is a *diameter* $\Delta$ such that any state can be reached from any other state in at most $\Delta$ transitions. For instance, a fully connected Markov chain pertains to $\Delta = 1$. Note that having $\Delta > 1$ is often natural, e.g., two successive punctuation marks (such as semicolons) are unlikely in an English document. When the transition distributions do not change with time $i$, the model is called *time-homogeneous*, otherwise it is *non-stationary*, *time-varying* or *non-homogeneous* [12, 13, 14]. Given a sequence $\boldsymbol{w}$ of $T$ observations, the decoding problem is to infer a most probable sequence or *path* $\mathbf{y}^*$ of $T$ states

$$\mathbf{y}^* \;\in\; \operatorname*{argmax}_{\mathbf{y}} \mathbb{P}(\mathbf{y},\boldsymbol{w}) \;=\; \operatorname*{argmax}_{\mathbf{y}} \log \mathbb{P}(\mathbf{y},\boldsymbol{w}) \;.$$

| Model | Reward $\overline{R}_i(y_i\|y_{[i-n,i-1]})$ | Model | Reward $\overline{R}_i(y_i\|y_{[i-n,i-1]})$ |
|---|---|---|---|
| $(n+1)$-GRAM | $\log \mathbb{P}(y_i\|y_{i-n},\ldots,y_{i-1}) + \log \mathbb{P}(w_i\|y_i)$ | 1-HMM | $\log \mathbb{P}(y_i\|y_{i-1}) + \log \mathbb{P}(w_i\|y_i)$ |
| $n$-MEMM | $\log \dfrac{\exp(\boldsymbol{\theta}^\top \boldsymbol{\phi}(y_{[i-n,i-1]}, y_i, \boldsymbol{w}, i))}{\sum_{y_i'} \exp(\boldsymbol{\theta}^\top \boldsymbol{\phi}(y_{[i-n,i-1]}, y_i', \boldsymbol{w}, i))}$ | $n$-CRF | $\boldsymbol{\theta}^\top \boldsymbol{\phi}(y_{[i-n,i-1]}, y_i, \boldsymbol{w}, i)$ |

Table 1: Standard Markov models in the reward form. We use $y_{[i,j]}$ to denote $(y_i, y_{i+1}, \ldots, y_j)$.

Decoding is a key inference problem in other structured prediction settings [15, 16] as well, e.g., maximum entropy Markov models (MEMM) [17] and conditional random fields (CRF) [18, 19] employ learnable parameters $\boldsymbol{\theta}$ and define the conditional dependence of each state on the observations through feature functions $\boldsymbol{\phi}$. The decoding task in all these models can be expressed in the form

$$\mathbf{y}^* \in \operatorname*{argmax}_{\mathbf{y}} \sum_{i=1}^{T} \overline{R}_i(y_i|y_{i-n},\ldots,y_{i-1}),\tag{1}$$

where we have made the dependence on observations $\boldsymbol{w}$ implicit in the *reward* functions $\overline{R}_i$ as shown in Table 1. The Viterbi algorithm [1] is employed for solving problems of the form (1) exactly. However, the algorithm cannot decode any observation until it has processed the entire observation sequence, i.e., computed and stored for each state $s$ a most probable sequence of $T$ states that ends in $s$. We say an algorithm has a hard latency $L$ if $L$ is the smallest $B$ such that the algorithm needs to access at most $B + 1$ observations $w_i, w_{i+1}, \ldots, w_{i+B}$ to generate the label for observation $w_i$ at any time $i$ during the decoding process. Thus, the latency of Viterbi algorithm on a sequence of length $T$ is $T - 1$, which is prohibitive for large $T$, especially in memory impoverished systems such as IoT devices [20, 21, 22, 23]. Besides, the algorithm is not suitable for critical scenarios such as patient monitoring, intrusion detection, and credit card fraud monitoring where delay following the onset of a suspicious activity might be detrimental [24]. Moreover, low latency is desirable for tasks such as drug discovery that rely on detecting interleaved coding regions in massive gene sequences.

A lot of effort has been, and continues to be, invested into speeding up the Viterbi algorithm, or reducing its memory footprint [25]. Some prominent recent approaches include fast matrix multiplication [26], compression and storage reduction for HMM [27, 28, 29], and heuristics such as beam search and simulated annealing [30, 31]. Several of these methods are based on the observation that if all the candidate subsequences in the Viterbi algorithm converge at some point, then all subsequent states will share a common subsequence up to that point [32, 33]. However, these methods do not guarantee reduction in latency since, in the worst case, they still need to process all the rewards before producing any output. [24] introduced *Online Step Algorithm* (OSA), with provable guarantees, to handle *soft* latency requirements in first order models. However, OSA makes a strong assumption that uncertainty in any state label decreases with latency. This assumption does not hold for important applications such as genome data. Moreover, OSA does not provide a direct control over latency (which needs to be tuned), and is limited to first order fully connected settings. We draw inspiration from, and generalize, the work by [34] on online server allocation under what we view as first order fully connected non-homogeneous setting (when the number of servers is one).

**Our contributions**

We investigate the problem of online decoding with Markov chain models under hard latency constraints, and design almost optimal online deterministic and randomized algorithms for problems of the form (1). Our bounds apply to general settings, e.g., when the rewards vary with time (non-homogeneous settings), or even when they are presented in an adversarial or adaptive manner. Our guarantees hold for finite latency (i.e. not only asymptotically), and improve with increase in latency. Our algorithms are efficient dynamic programs that may not only be deployed in settings where Viterbi algorithm is typically used but also, as we mentioned earlier, several others where it is impractical. Thus, our work would potentially widen the scope of and expedite scientific discovery in several fields that rely critically on efficient online Markov decoding.

We also provide the first results on the limits of online Markov decoding under latency constraints. Specifically, we craft lower bounds for the online approximation of Viterbi algorithm in both deterministic and randomized ergodic chain settings. Moreover, we establish that no online algorithm can perform significantly better than our algorithms. In particular, our algorithms provide strong guarantees even for low latency, and nearly match the lower bounds for sufficiently large latency.

| | LOWER BOUND | UPPER BOUND (OUR ALGORITHMS) |
|---|---|---|
| DETERMINISTIC $(\Delta = 1, n = 1)$ | $1 + \dfrac{1}{L} + \dfrac{1}{L^2 + 1}$ | $\min\left\{\left(1 + \dfrac{1}{L}\right)\sqrt[L]{L+1}, 1 + \dfrac{4}{L-7}\right\}$ |
| RANDOMIZED $(\Delta = 1, n = 1, \epsilon > 0)$ | $1 + \dfrac{(1-\epsilon)}{L+\epsilon}$ | $1 + \dfrac{1}{L}$ |
| DETERMINISTIC | $1 + \dfrac{\tilde{\Delta}}{L}\left(1 + \dfrac{\tilde{\Delta} + L - 1}{(L - \tilde{\Delta} - 1)^2 + 4\tilde{\Delta}L - 3\tilde{\Delta}}\right)$ | $1 + \min\left\{\Theta\left(\dfrac{\log L}{L - \tilde{\Delta} + 1}\right),\right.$ $\left.\Theta\left(\dfrac{1}{L - 8\tilde{\Delta} + 1}\right)\right\}$ |
| RANDOMIZED $(\epsilon > 0)$ | $1 + \dfrac{\left(2^{\Delta-1}\lceil 1/\epsilon\rceil - 1\right)n}{2^{\Delta-1}\lceil 1/\epsilon\rceil L + n}$ | $1 + \Theta\left(\dfrac{1}{L - \tilde{\Delta} + 1}\right)$ |

Table 2: Summary of our results in terms of the competitive ratio $\rho$. Note that the effective diameter $\tilde{\Delta} = \Delta + n - 1$. To fit some results within the margins, we use the standard notation $\Theta(\cdot)$ on the growth of functions and summarize the performance of Peek Search asymptotically in $L$. The non-asymptotic dependence on $L$ is made precise in all cases in our theorem statements.

We introduce several novel ideas and analyses in the context of approximate Markov decoding. For example, we approximate a non-discounted objective over *horizon* $T$ by a sequence of smaller discounted subproblems over horizon $L + 1$, and track down the Viterbi algorithm by essentially foregoing rewards on at most $\tilde{\Delta} = \Delta + n - 1$ steps in each smaller problem. Our design of constructions toward proving lower bounds in a setting predicated on interplay of several heterogeneous variables, namely $L$, $n$, and $\Delta$, is another significant technical contribution. We believe our tools will foster designing new online algorithms, and establishing combinatorial bounds for related settings such as dynamic Bayesian networks, hidden semi-Markov models, and model based reinforcement learning.

## 2 Overview of our results

We introduce some notation. We define $[a, b] \triangleq (a, a+1, \ldots, b)$ and $[N] \triangleq (1, 2, \ldots, N)$. Likewise, $y_{[N]} \triangleq (y_1, \ldots, y_N)$ and $y_{[a,b]} \triangleq (y_a, \ldots, y_b)$. We denote the last $n$ states visited by the online algorithm at time $i$ by $\hat{y}_{[i-n,i-1]}$, and those by the optimal offline algorithm by $y^*_{[i-n,i-1]}$. Defining positive reward functions $R_i = \overline{R}_i + p$ by adding a sufficiently large positive number $p$ to each reward, we note from (1) that an optimal sequence of states for input observations $w$ of length $T$ is

$$y^*_{[1,T]} \in \arg\max_{y_1,\ldots,y_T} \sum_{i=1}^{T} R_i(y_i | y_{[i-n,i-1]}) \ . \tag{2}$$

We use $OPT$ to denote the total reward accumulated by the optimal offline algorithm, and $ON$ to denote that received by the online algorithm. We evaluate the performance of any online algorithm in terms of its *competitive ratio* $\rho$, which is defined as the ratio $OPT/ON$. That is,

$$\rho = \sum_{i=1}^{T} R_i(y^*_i | y^*_{[i-n,i-1]}) \bigg/ \sum_{i=1}^{T} R_i(\hat{y}_i | \hat{y}_{[i-n,i-1]}) \ .$$

Clearly, $\rho \geq 1$. Our goal is to design online algorithms that have competitive ratio close to 1. For randomized algorithms, we analyze the ratio obtained by taking expectation of the total online reward over its internal randomness. The performance of any online algorithm depends on the order $n$, latency $L$, and *diameter* $\Delta$. Table 2 provides a summary of our results. Note that our algorithms are asymptotically optimal in $L$. For the finite $L$ case, we first consider the fully connected first order models. Our randomized algorithm matches the lower bound even[1] with $L = 1$ since we may set $\epsilon$ arbitrarily close to 0. Note that just with $L = 1$, our deterministic algorithm achieves a competitive

ratio 4, and this ratio reduces further as $L$ increases. Moreover our ratio rapidly approaches the lower bound with increase in $L$. Finally, in the general setting, our algorithms are almost optimal when $L$ is sufficiently large compared to $\tilde{\Delta} = \Delta + n - 1$. We call $\tilde{\Delta}$ the *effective diameter* since it nicely encapsulates the roles of order $n$ and diameter $\Delta$ toward the quality of approximation.

The rest of the paper is organized as follows. We first introduce and analyze our deterministic *Peek Search* algorithm for homogeneous settings in section 3. We then introduce the *Randomized Peek Search* algorithm in section 4. In section 5, we propose the deterministic *Peek Reset* algorithm that performs better than deterministic Peek Search for large $L$. We then present the lower bounds in section 6, and demonstrate the merits of our approach via strong empirical evidence in section 7. We analyze the non-homogeneous setting, and provide all the proofs in the supplementary material.

## 3 Peek Search

Our idea is to approximate the sum of rewards over $T$ steps in (2) by a sequence of smaller problems over $L + 1$ steps. The Peek Search algorithm is so named since at each time $i$, besides the observation $w_i$, it *peeks* into the next $L$ observations $w_{i+1}, \ldots, w_{i+L}$. The algorithm then leverages the sub-sequence $w_{[i,i+L]}$ to decide its next state $\hat{y}_i$. Let $\gamma \in (0,1)$ be a *discount factor*. Specifically, the algorithm repeats the following procedure at each time $i$. First, it finds a path of length $L + 1$ emanating from the current state $\hat{y}_{i-1}$ that fetches maximum *discounted* reward. The discounted reward on any path is computed by scaling down the $\ell^{th}$ edge, $\ell \in \{0, \ldots, L\}$, on the path by $\gamma^\ell$. Then, the algorithm moves to the first state of this path and repeats the procedure at time $i + 1$. Note that at time $i + 1$, the algorithm need not continue with the second edge on the optimal discounted path computed at previous time step $i$, and is free to choose an alternative path. Formally, at time $i$, the algorithm computes $\tilde{y}_i \triangleq (\tilde{y}_i^0, \tilde{y}_i^1, \ldots, \tilde{y}_i^L)$ that maximizes the following objective over valid paths $y = (y_i, \ldots, y_{i+L})$,

$$R(y_i | \hat{y}_{[i-n,i-1]}) \; + \; \sum_{j=1}^{n-1} \gamma^j R(y_{i+j} | \hat{y}_{[i-n+j,i-1]}, y_{[i,i+j-1]}) \; + \; \sum_{j=n}^{L} \gamma^j R(y_{i+j} | y_{[i+j-n,i+j-1]}) \,,$$

sets the next state $\hat{y}_i = \tilde{y}_i^0$, and receives the reward $R(\hat{y}_i | \hat{y}_{[i-n,i-1]})$. Note that we have dropped the subscript $i$ from $R_i$ since in the homogeneous settings, the reward functions do not change with time $i$. For any given $L$ and $\tilde{\Delta}$, we optimize to get the optimal $\gamma$. Intuitively, $\gamma$ may be viewed as an *explore-exploit* parameter that indicates the confidence of the online algorithm in the best discounted path: $\gamma$ grows as $L$ increases, and thus a high value of $\gamma$ indicates that the path computed at a time $i$ may be worth tracing at subsequent few steps as well. In contrast, the algorithm is uncertain for small values of $L$. We have the following near-optimal result on the performance of Peek Search.

**Theorem 1.** *The competitive ratio of Peek Search on Markov chain models of order $n$ with diameter $\Delta$ for $L \geq \Delta + n - 1$ is $\rho \leq (\gamma^{\Delta+n-1} - \gamma^{L+1})^{-1}$. Setting $\gamma = \sqrt[(L-\Delta-n+2)]{\dfrac{\Delta+n-1}{L+1}}$, we get*

$$\rho \;\; \leq \;\; \frac{L+1}{L-\Delta-n+2} \left( \frac{L+1}{\Delta+n-1} \right)^{(n+\Delta-1)/(L-\Delta-n+2)} \;\; = \;\; 1 + \Theta\left( \frac{\log L}{L - \tilde{\Delta} + 1} \right) \,.$$

*Proof.* (Sketch) We first consider the fully connected first order setting (i.e. $n = 1, \Delta = 1$). Our analysis hinges on two important facts. Since Peek Search chooses a path that maximizes the total discounted reward over next $(L + 1)$ steps, it is guaranteed to fetch all of the discounted reward pertaining to the optimal path except that available on the first step of the optimal path (see Fig. 1 for visual intuition). Alternatively, Peek Search could have persisted with the maximizing path computed at the previous time step (recall that only first step of this path was taken to reach the current state). We exploit the fact that this path is now worth $1/\gamma$ times its anticipated value at the previous step.

Now consider $n > 1$. The online algorithm may jump to any state on the optimal offline, i.e. Viterbi path, in one step. However, the reward now depends on the previous $n$ states, and so the online algorithm may have to wait additional $n - 1$ steps before it could trace the subsequent optimal path. Finally, as explained in Fig. 1, when $\Delta > 1$, the online algorithm may have to forfeit rewards on (at most) $\Delta$ steps, in addition to the $n - 1$ steps, in order to join the optimal path. $\qquad\square$

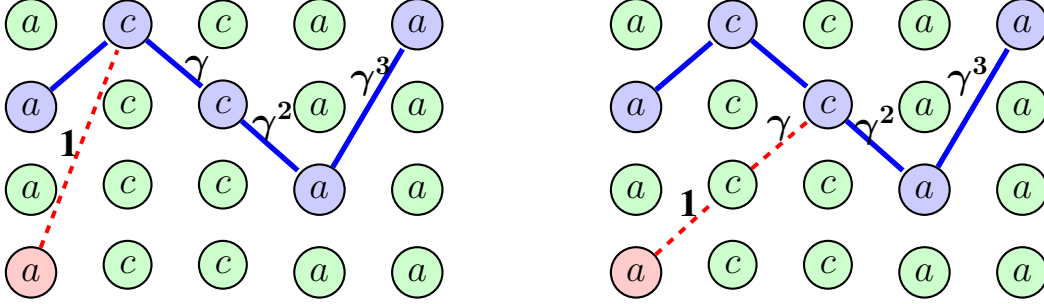

Figure 1: **Visual intuition for the setting** $n = 1$. (**Left**) A trellis diagram obtained by unrolling a fully connected Markov graph (i.e. diameter $\Delta = 1$). The states are shown along the rows, and time along the columns. The system is currently in state 4 (shown in red), and has access to rewards and observations (shown inside circles) for the next $(L + 1)$ steps. The unknown optimal path is shown in blue, and the weights with which rewards are scaled are shown on the edges. One option available with the online algorithm is to jump to state 1 (possibly fetching zero reward) and then follow the optimal path for the subsequent $L$ steps. Note that the online algorithm might choose a different path, but it is guaranteed at least as much reward since it maximizes the discounted reward over $L + 1$ steps. $\gamma$ approaches 1 with increase in $L$. This would ensure that the online algorithm makes nearly the most of $L$ steps every $L + 1$ steps. (**Right**) If the graph is not fully connected, some of the transitions may not be available (e.g. state 4 to state 1 in our case). Therefore, the online algorithm might not be able to join the optimal path in one step, and thus may have to forgo additional rewards.

We show in the supplementary material that this guarantee on the performance of Peek Search extends to the non-homogeneous settings, including those where the rewards may be adversarially chosen. Note that naïvely computing a best path by enumerating all paths of length $L + 1$ would be computationally prohibitive since the number of such paths is exponential in $L$. Fortunately, we can design an efficient dynamic program for Peek Search. Specifically, we can show that for every $\ell \in \{1, 2, \ldots, L\}$, the reward on the optimal discounted path of length $\ell$ can be recursively computed from an optimal path of length $\ell$-1 using $O(|K|^n)$ computations. We have the following result.

**Theorem 2.** *Peek Search can compute a best $\gamma$-discounted path for the next $L + 1$ steps, in $n^{th}$ order Markov chain models, in time $O(L|K|^n)$, where $K$ is the set of states.*

We outline an efficient procedure, underlying Theorem 2, in the supplementary material. We now introduce two algorithms that do not recompute the paths at each time step. These algorithms provide even tighter (expected) approximation guarantees than Peek Search for larger values of the latency $L$.

## 4    Randomized Peek Search

We first introduce the *Randomized Peek Search* algorithm, which removes the asymptotic log factor from the competitive ratio in Theorem 1. Unlike Peek Search, this method does not discount the rewards on paths. Specifically, the algorithm first selects a *reset point* $\ell$ uniformly at random from $\{1, 2, \ldots, L + 1\}$. This number is a private information for the online algorithm. The randomized algorithm recomputes the optimal *non-discounted* path (which corresponds to $\gamma = 1$) of length $(L + 1)$, once every $L + 1$ steps, at each time $i * (L + 1) + \ell$, and follows this path for next $L + 1$ steps without any updates. We have the following result that underscores the benefits of randomization.

**Theorem 3.** *Randomized Peek Search achieves, in expectation, on Markov chain models of order $n$ with diameter $\Delta$ a competitive ratio*

$$ \rho \quad \leq \quad 1 + \frac{\Delta + n - 1}{L + 1 - (\Delta + n - 1)} \quad = \quad 1 + \Theta\left(\frac{1}{L - \tilde{\Delta} + 1}\right) . $$

*Proof.* (Sketch) Since it maximizes the non-discounted reward, for each random reset point $\ell$, the online algorithm receives at least as much reward as the optimal offline algorithm minus the reward on at most $\tilde{\Delta}$ steps every $L + 1$ steps. We show that, in expectation, Peek Reset misses on only (at most) a $\tilde{\Delta}/(L + 1)$ fraction of the optimal offline reward. □

Theorem 2 is essentially tight since it nearly matches the lower bound as described previously in section 2. We leverage insights from Randomized Peek Search to translate its almost optimal expected performance to the deterministic setting. Specifically, we introduce the Peek Reset algorithm that may be loosely viewed as a *derandomization* of Randomized Peek Search. The main trick is to conjure a sequence of reset points, each over a variable number of steps. This allows the algorithm to make adaptive decisions about when to forgo rewards. Both Randomized Peek Search and Peek Reset can compute rewards on their paths efficiently by using the procedure for Peek Search as a subroutine.

## 5   Peek Reset

We now present the deterministic *Peek Reset* algorithm that performs better than Peek Search when the latency $L$ is sufficiently large. Like Randomized Peek Search, Peek Reset recomputes a best non-discounted path and takes multiple steps on this path. However, the number of steps taken is not fixed to $L + 1$ but may vary in each *phase*. Specifically, let $(i)$ denote the time at which phase $i$ begins. The algorithm follows, in phase $i$, a sequence of states $\hat{y}(i) \triangleq (\hat{y}_{(i)}, \hat{y}_{(i)+1}, \ldots, \hat{y}_{T_i-1})$ that maximizes the following objective over valid paths $y = (y_{(i)}, \ldots, y_{T_i-1})$ :

$$
\begin{aligned}
f(y) \quad \triangleq \quad & R(y_{(i)}|\hat{y}_{[(i)-n,(i)-1]}) \quad + \quad \sum_{j=1}^{n-1} R(y_{(i)+j}|\hat{y}_{[(i)-n+j,(i)-1]}, y_{[(i),(i)+j-1]}) \\
& \quad + \quad \sum_{j=n}^{T_i-(i)-1} R(y_{(i)+j}|y_{[(i)+j-n,(i)+j-1]}) \ ,
\end{aligned}
$$

where $T_i$ is chosen from the following set (breaking ties arbitrarily)

$$
\arg\min_{t \in [(i)+L/2+1, (i)+L]} \max_{(y_{t-n},\ldots,y_t)} R(y_t|y_{[t-n,t-1]}) \ .
$$

Then, the next phase $(i + 1)$ begins at time $T_i$. We have the following result.

**Theorem 4.** *The competitive ratio of Peek Reset on Markov chain models of order $n$ with diameter $\Delta$ for latency $L$ is*

$$
\rho \quad \leq \quad 1 + \frac{2(\Delta + n)(\Delta + n - 1)}{L - 8(\Delta + n - 1) + 1} \quad = \quad 1 + \Theta\left(\frac{1}{L - 8\tilde{\Delta} + 1}\right) \ .
$$

*Proof.* (Sketch) The algorithm gives up reward on at most $\tilde{\Delta}$ steps every $L + 1$ steps, however these steps are cleverly selected. Note that $T_i$ is chosen from the interval $[(i) + L/2 + 1, (i) + L]$, which contains steps from both phases $(i)$ and $(i + 1)$. Thus, the algorithm gets to peek into phase $(i + 1)$ before deciding on the number of steps to be taken in phase $(i)$. $\qquad\square$

A comparison of Theorem 4 with Theorem 1 reveals that Peek Reset provides better upper bounds on the approximation quality than Peek Search for sufficiently large latency. In particular, for the fully connected first order setting, i.e. $\tilde{\Delta} = 1$, the competitive ratio of Peek Reset is at most $1 + 4/(L - 7)$ which is better than the corresponding worst case bound for Peek Search when $L \geq 50$. Thus, Peek Search is better suited for applications with severe latency constraints whereas Peek Reset may be preferred in less critical scenarios. We now establish that no algorithm, whether deterministic or randomized, can provide significantly better guarantees than our algorithms under latency constraints.

## 6   Lower Bounds

We now state our lower bounds on the performance of any deterministic and any randomized algorithm in the general non-homogeneous ergodic Markov chain models. The proofs revolve around our novel $\Delta$-dimensional prismatic polytope constructions, where each vertex corresponds to a state. We disentangle the interplay between $L$, $\Delta$, and $n$ (see Fig. 2 for visual intuition).

**Theorem 5.** *The competitive ratio of any deterministic online algorithm on $n^{th}$ order (time-varying) Markov chain models with diameter $\Delta$ for latency $L$ is greater than*

$$
1 + \frac{\tilde{\Delta}}{L}\left(1 + \frac{\tilde{\Delta} + L - 1}{(\tilde{\Delta} + L - 1)^2 + \tilde{\Delta}}\right) \ .
$$

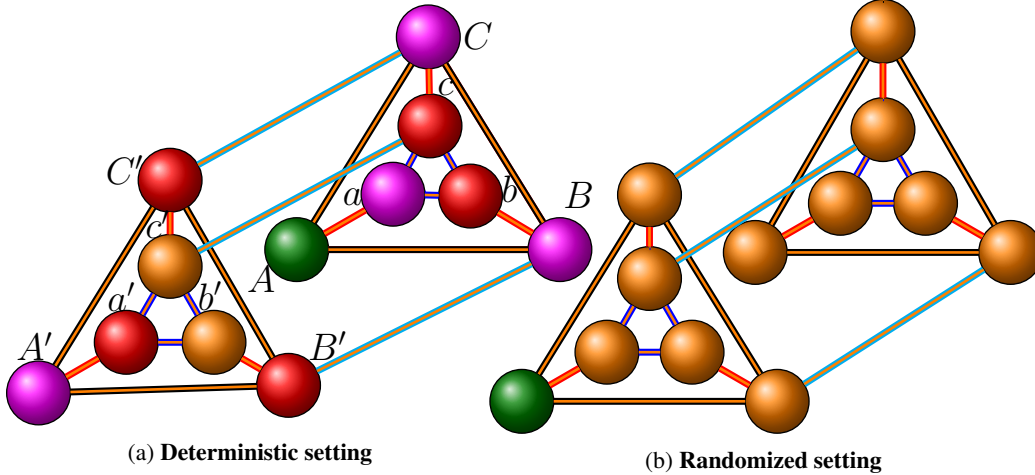

|                              |                           |
|:----------------------------:|:-------------------------:|
| (a) **Deterministic setting** | (b) **Randomized setting** |

Figure 2: **Constructions for lower bounds with $\Delta = 3$. (Left)** $ABC$ and $abc$ are opposite faces of a triangular prism, and $A'B'C'$ and $a'b'c'$ are their translations. The resulting prismatic polytope has the property that distance between the farthest vertices is $\Delta$. Different colors are used for edges on different faces, and same color for translated faces to aid visualization (we have also omitted some edges that connect faces to their translated faces, in order to avoid clutter). A priori the rewards for the $L + 1$ steps are same across all vertices (i.e. states). Thus, due to symmetry of the polytope, the online algorithm arbitrarily chooses some vertex (shown here in green). The states that can be reached via shortest paths of same length from this vertex are displayed in same color (magenta, red, or orange). The adversary reveals the rewards for an additional, i.e. $(L + 2)^{th}$, time step such that states at distance $d \in [\Delta]$ from the green state would fetch $(n + d - 1)\alpha$ for some $\alpha$, while the green state would yield 0. Under the Markov dependency rule that a state yields reward only if it has been visited $n$ consecutive times, the online algorithm fails to obtain any reward in the $(L + 2)^{th}$ step regardless of the state sequence it traces. The optimal algorithm, due to prescience, gets the maximum possible reward $(n + \Delta - 1)\alpha$ for this step. **(Right)** In the randomized setting, all states fetch zero reward at the final step except a randomly chosen state (shown in green) that yields reward $n$. The probability that the randomized online algorithm correctly guesses the green state at the initial time step is exponentially small in $\Delta$. In all other cases, it must forgo this reward, and thus its expected reward is low compared to the optimal algorithm for large $\Delta$.

*In particular, when $n = 1$, $\Delta = 1$, the ratio is larger than $1 + \dfrac{1}{L} + \dfrac{1}{L^2 + 1}$ .*

**Theorem 6.** *For any $\epsilon > 0$, the competitive ratio of any randomized online algorithm, that is allowed latency L, on $n^{th}$ order (time-varying) Markov chain models with $\Delta = 1$ is at least $1 + \dfrac{(1 - \epsilon)n}{L + \epsilon n}$ .*
*For a general diameter $\Delta$, the competitive ratio is at least $1 + \dfrac{\left(2^{\Delta-1}\lceil 1/\epsilon \rceil - 1\right)n}{2^{\Delta-1}\lceil 1/\epsilon \rceil L + n}$ .*

We now analyze the performance of our algorithms in the wake of these lower bounds. Note that when $\tilde{\Delta} = 1$, Randomized Peek Search (Theorem 3) matches the lower bound in Theorem 6 even with $L = 1$, since we may set $\epsilon$ arbitrarily close to 0. Similarly, in the deterministic setting, Peek Search achieves a competitive ratio of 4 with $L = 1$ (Theorem 1), which is within twice the theoretically best possible performance (i.e. a ratio of 2.5) as specified by Theorem 5. Moreover, the performance approaches the lower bound with increase in $L$ as Peek Reset takes center stage (Theorem 4). In the general setting, our algorithms are almost optimal when $L$ is sufficiently large compared to $\tilde{\Delta}$.

Note that we do not make any distributional assumptions on the rewards for any $(L + 1)$-long peek window. Thus, our algorithms accommodate various settings, including those where the rewards may be revealed in an adaptive (e.g. non-stochastic, possibly adversarial) manner.

We now proceed to our experiments that accentuate the practical implications of our work.

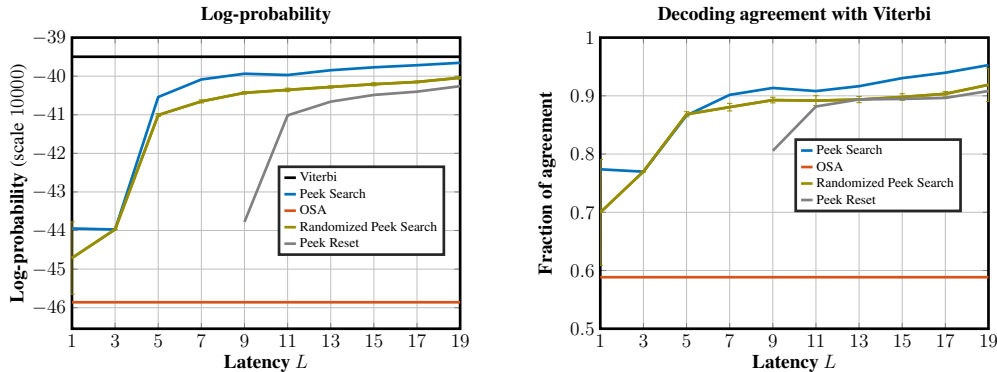

Figure 3: **Evaluation of performance on genome sequence data**. The data consists of 73385 sites, each of which is to be labeled with one of the four states. The log-probability values on the right have been scaled down by a factor of $10^4$ to avoid clutter near the vertical axis. Peek Search achieves almost optimal performance with a latency of only about 20, which is over three orders of magnitude less than the optimal Viterbi algorithm. The corresponding predictions agreed with the Viterbi algorithm on more than 95% of all sites. Peek Reset and Randomized Peek Search also performed well especially for larger values of $L$. In contrast, OSA was found to be significantly suboptimal.

## 7 Experiments

We describe the results of our experiments on two real datasets. We first compare the performance of our methods with the state-of-the-art Online Step Algorithm (OSA) [24] that also provides theoretical guarantees for first order Markov decoding under latency constraints. OSA hinges on a strong assumption that uncertainty in any state label decreases with increase in latency. We found that this assumption does not hold in the context of an important application, namely, genome decoding. In contrast, since our algorithms do not make any such assumptions, they achieve much better performance as expected. Furthermore, unlike our algorithms, OSA does not provide a direct control over the latency $L$. Specifically, OSA relies on a hyperparameter $\lambda$, that may require extensive tuning, to achieve a good trade-off between latency and accuracy. Our empirical findings thus underscore the promise of our algorithms toward expediting scientific progress in fields like drug discovery. We then demonstrate that Peek Search performs exceptionally well on the task of part-of-speech tagging on the Brown corpus data even for $L = 1$. We also provide evidence that heuristics such as Beam Search can be adapted to approximate optimal discounted paths efficiently within peek windows of length $(L + 1)$. This computational benefit, however, comes at the expense of theoretical guarantees.

### 7.1 Genome sequencing

We experimented with the Glycerol TraSH genome data [35] pertaining to M. tuberculosis transposon mutants. Our task was to label each of the 73385 gene sites with one of the four states, namely essential (ES), growth-defect (GD), non-essential (NE), and growth-advantage (GA). These states represent different categories of gene *essentiality* depending on their read-counts (i.e. emissions), and the labeling task is crucial toward identifying potential drug targets for antimicrobial treatment [35]. We used the parameter settings suggested by [35] for decoding with an HMM.

Note that for this problem, the Viterbi algorithm and heuristics such as beam search need to compute the optimal paths of length equal to the number of sites, i.e. in excess of 73000, thereby incurring very high latency. However, as Fig. 3 shows, Peek Search achieved near-optimal log-probability (the Viterbi objective in (1)) with a latency of only about 20, which is less than that of Viterbi by a factor in excess of 3500. Moreover, the state sequence output by Peek Search agreed with the Viterbi labels on more than 95% of the sites. We observe that, barring downward blips from $L = 1$ to $L = 3$ and from $L = 9$ to $L = 11$, the performance improved with $L$. As expected, for all $L$, including those featuring in the blips, the log-probability values were verified to be consistent with our theoretical guarantees. On the other hand, we found OSA to be significantly suboptimal in terms of both log-probability and label agreement. In particular, OSA agreed with the optimal algorithm (Viterbi) on only 58.8% of predictions under both entropy and expected classification error

| Latency | Method | Log-probability | Tagging accuracy (%) |
|---|---|---|---|
| | Viterbi | -117.29 +/- .53 | 97.4 +/- .02 |
| $L = 1$ | Peek Search | -117.40 +/- .54 | 97.0 +/- .01 |
| $L = 1$ | Approximate Peek Search (3 beams) | -117.40 +/- .54 | 97.0 +/- .02 |
| $L = 2$ | Peek Search | -117.34 +/- .54 | 97.2 +/- .01 |
| $L = 2$ | Approximate Peek Search (3 beams) | -117.34 +/- .54 | 97.2 +/- .01 |
| $L = 3$ | Peek Search | -117.33 +/- .54 | 97.3 +/- .02 |
| $L = 3$ | Approximate Peek Search (3 beams) | -117.33 +/- .54 | 97.3 +/- .02 |

Table 3: Part-of-speech tagging on Brown data.

measures suggested in [24]. In contrast, just with $L = 1$, Peek Search matched with Viterbi on $77.4\%$ predictions thereby outperforming OSA by an overwhelming amount (over $30\%$). We varied the OSA hyperparameter $\lambda \in \{10^{-4}, 10^{-1}, \ldots, 10^4\}$ under both the entropy and the expected classification error measures suggested by [24] to tune for $L$ (as noted in [24], large values of $\lambda$ penalize latency). However, the performance of OSA (as shown in Fig. 3) did not show any improvement.

Fig. 3 also shows the performance of Randomized Peek Search (averaged over 10 independent runs) and Peek Reset. Since the guarantees of Peek Reset are meaningful for $L$ large enough (exceeding 7), we show results with Peek Reset for $L \geq 9$. Both these methods were found to be better than OSA on the genome data. Moreover, as expected, the performance of Peek Reset improved with increase in $L$. In particular, the scaled log-probabilities under Peek Reset for $L = 50$ and $L = 100$ were observed, respectively, to be -39.69 and -39.56. Moreover, the decoded sequences agreed with Viterbi on $97.32\%$ and $98.68\%$ of the sites respectively. For smaller values of $L$, Peek Search turned out to be better than both Peek Reset and Randomized Peek Search.

We also evaluated the performance of Beam Search. Note that despite efficient greedy path expansion, Beam Search with $k$ beams (BS-$k$) has high latency (same as Viterbi) since no labels can be generated until the $k$-greedy paths are computed for entire sequence and backpointers are traced back to the start. We found that BS-2 performed worse than Peek Search for $L \geq 5$. Also, BS-3 recorded log prob.-39.61 and decoding agreement $97.73\%$ (worse than Peek Search with $L = 50$). BS-$k$ matched the Viterbi performance for $k \geq 4$.

Finally, note that instead of choosing $\gamma$ optimally, one could fix $\gamma$ to some other value in Peek Search. In particular, setting $\gamma = 1$ amounts to having Peek Search move one step on a path with maximum non-discounted reward during each peek window. We found that Peek Search with $\gamma = 1$ obtained a sub-optimal scaled log-probability of -45.86 (and $58.8\%$ decoding match with Viterbi). However, setting $\gamma$ optimally did not make any difference for larger $L$.

## 7.2 Part-of-speech tagging

For our second task, we focused on the problem of decoding the part-of-speech (POS) tags for sentences in the standard Brown corpus data. The corpus comprises 57340 sentences of different lengths that have 1161192 tokens in total. The corpus is not divided into separate train and test sets. Therefore, we formed 5 random partitions each having 80% train and 20% test sentences. The train set was used to estimate the parameters of a first order HMM, and these parameters were then used to predict the tags for tokens in the test sentences. For each test sentence that had all its tokens observed in the train data, we computed its log-probability using its predicted tags (note that the Viterbi algorithm maximizes this quantity in (1) exactly).

We computed the average log-probability over these test sentences for both the Viterbi algorithm, and Peek Search for different values of latency $L$. We also computed the accuracy of tag predictions, i.e. the fraction of test tokens whose predicted tags matched their ground truth labels. We report the results (averaged over 5 independent train-test partitions) in Table 3. We observed that Peek Search nearly matched the performance of Viterbi.[2] Moreover, similar results were obtained when we used 3 beams to approximate the optimal $\gamma$-discounted reward within each $(L + 1)$-long peek window. Thus, we can potentially design fast yet accurate heuristics for some low latency settings.

## Footnotes

[1] It is easy to construct examples where any algorithm with no latency may be made to incur an arbitrarily high $\rho$. Thus, in the fully connected first order Markov setting, online learning is meaningful only for $L \geq 1$.

[2]We found that OSA also achieved almost optimal performance on the Brown corpus.

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
