[Supplementary Material]

## Supplementary Material

We now provide detailed proofs of all the theorems stated in the main text.

For improved readability, instead of proving Theorem 1 immediately, we start with two simpler settings, namely, (a) first order fully connected,[3] and (b) $n^{th}$ order fully connected. Together with Theorem 1, these results will help segregate the effect of $n$ from that of $\Delta$ on the competitive ratio.

## A  First order chain models with $\Delta = 1$

**Lemma 1.** *The competitive ratio of Peek Search on first order Markov chain models with $\Delta = 1$ for $L \geq 1$ is*

$$\rho \;\; \leq \;\; \left(1 + \frac{1}{L}\right) \sqrt[L]{L+1} \, .$$

*Proof.* Recall that at each time step $i$, our online algorithm solves the following optimization problem over variables $y \triangleq (y_i, y_{i+1}, \ldots, y_{i+L}) \in S(i, L)$, i.e. the set of valid paths of length $L + 1$ that emanate from the state at time $i$:

$$M_i = \arg\max_{y \in S(i,L)} R\left(y_i | \hat{y}_{i-1}\right) + \sum_{j=1}^{L} \gamma^j R(y_{i+j} | y_{i+j-1}).$$

Note that the set $M_i$ may have more than one path that maximizes the discounted sum. Breaking ties arbitrarily, let the online algorithm choose $\tilde{y}_i \triangleq (\hat{y}_i, \tilde{y}_i^1, \ldots, \tilde{y}_i^L) \in M_i$ (and reach the state $\hat{y}_i$). Let $\{y_t^* \mid t \in [T]\}$ be the optimal path over the entire horizon. Since $\Delta = 1$, one of the candidate paths considered by the online algorithm is the optimal segment $(y_i^*, y_{i+1}^*, \ldots, y_{i+L}^*)$. Since $\tilde{y}_i \in M_i$, we must have

$$R(\hat{y}_i | \hat{y}_{i-1}) \; + \; \gamma R(\tilde{y}_i^1 | \hat{y}_i) \; + \; \sum_{j=2}^{L} \gamma^j R(\tilde{y}_i^j | \tilde{y}_i^{j-1})$$

$$\geq \quad R(y_i^* | \hat{y}_{i-1}) \; + \; \sum_{j=1}^{L} \gamma^j R(y_{i+j}^* | y_{i+j-1}^*)$$

$$\geq \quad \sum_{j=1}^{L} \gamma^j R(y_{i+j}^* | y_{i+j-1}^*) \, , \tag{3}$$

where the last inequality follows since all rewards are non-negative, and thus in particular, $R(y_i^* | \hat{y}_{i-1}) \geq 0$.

An alternate path considered by the online algorithm is $(\tilde{y}_{i-1}^1, \ldots, \tilde{y}_{i-1}^L, \bar{y}_{i-1}^{L+1})$, where $(\tilde{y}_{i-1}^1, \ldots, \tilde{y}_{i-1}^L)$ are the last $L$ steps of the path $\tilde{y}_{i-1} \in M_{i-1}$ (i.e. the path chosen at time $i - 1$) and $\bar{y}_{i-1}^{L+1}$ is an arbitrary valid transition from state $\tilde{y}_{i-1}^L$. Again since this transition fetches a non-negative reward, we must have

$$R(\hat{y}_i | \hat{y}_{i-1}) \; + \; \gamma R(\tilde{y}_i^1 | \hat{y}_i) \; + \; \sum_{j=2}^{L} \gamma^j R(\tilde{y}_i^j | \tilde{y}_i^{j-1})$$

$$\geq \quad R(\tilde{y}_{i-1}^1 | \hat{y}_{i-1}) \; + \; \sum_{j=1}^{L-1} \gamma^j R(\tilde{y}_{i-1}^{j+1} | \tilde{y}_{i-1}^j) \, . \tag{4}$$

Multiplying (3) by $1 - \gamma$ and (4) by $\gamma$, and adding the resulting inequalities, we get

$$R(\hat{y}_i|\hat{y}_{i-1}) + \gamma R(\tilde{y}_i^1|\hat{y}_i) + \sum_{j=2}^{L}\gamma^j R(\tilde{y}_i^j|\tilde{y}_i^{j-1})$$

$$\geq \sum_{j=1}^{L}(1-\gamma)\gamma^j R(y_{i+j}^*|y_{i+j-1}^*) + R(\tilde{y}_{i-1}^1|\hat{y}_{i-1}) + \sum_{j=1}^{L-1}\gamma^{j+1}R(\tilde{y}_{i-1}^{j+1}|\tilde{y}_{i-1}^j)$$

$$= \sum_{j=1}^{L}(1-\gamma)\gamma^j R(y_{i+j}^*|y_{i+j-1}^*) + \gamma R(\tilde{y}_{i-1}^1|\hat{y}_{i-1}) + \sum_{k=2}^{L}\gamma^k R(\tilde{y}_{i-1}^k|\tilde{y}_{i-1}^{k-1}),$$

where the last inequality follows from a change of variable, namely, $k = j + 1$. Summing across all time steps $i$,

$$\sum_i R(\hat{y}_i|\hat{y}_{i-1}) + \underbrace{\sum_i\left(\gamma R(\tilde{y}_i^1|\hat{y}_i) + \sum_{j=2}^{L}\gamma^j R(\tilde{y}_i^j|\tilde{y}_i^{j-1})\right)}_{DR1}$$

$$\geq \sum_i\sum_{j=1}^{L}(1-\gamma)\gamma^j R(y_{i+j}^*|y_{i+j-1}^*) + \underbrace{\sum_i\left(\gamma R(\tilde{y}_{i-1}^1|\hat{y}_{i-1}) + \sum_{j=2}^{L}\gamma^j R(\tilde{y}_{i-1}^j|\tilde{y}_{i-1}^{j-1})\right)}_{DR2}.$$

Without loss of generality, we can assume that all transitions between states in the first $L + 1$ time steps and the last $L + 1$ steps fetch zero reward.[4] Now note that both $DR1$ and $DR2$ consist of terms that pertain to leftover discounted rewards on optimal $(L + 1)$-paths computed by Peek Search (recall we take only the first step on each such path). In fact, the terms are common to both sides except for those that fall in $(L + 1)$-length windows at the beginning or the end. Since first and last $(L + 1)$ steps fetch zero reward, we can safely disregard these windows. Thus, by telescoping over $i$, we have

$$\sum_i R(\hat{y}_i|\hat{y}_{i-1}) \geq \sum_i\sum_{j=1}^{L}(1-\gamma)\gamma^j R(y_{i+j}^*|y_{i+j-1}^*).$$

Defining a variable $s = i + j$, and interchanging the two sums, we note that the right side becomes

$$(1-\gamma)\sum_{j=1}^{L}\gamma^j\sum_s R(y_s^*|y_{s-1}^*).$$

That is, every reward subsequent to $L + 1$ steps appears with discounts $\gamma, \gamma^2, \ldots, \gamma^L$. Summing the geometric series, we note that the ratio of the total reward obtained by the optimal offline algorithm to that by the online algorithm, i.e. the competitive ratio $\rho$ is at most $\gamma^{-1}(1-\gamma^L)^{-1}$. The result follows by setting $\gamma = \sqrt[L]{1/(L+1)}$. $\qquad\square$

# B $\quad n^{th}$ order chain models with $\Delta = 1$

**Lemma 2.** *The competitive ratio of Peek Search on Markov chain models of order $n$ with $\Delta = 1$ for $L \geq n$ is*

$$\rho \leq \frac{L+1}{L-n+1}\left(\frac{L+1}{n}\right)^{n/(L-n+1)} = 1 + \Theta\left(\frac{\log L}{L-n+1}\right).$$

*Proof.* For $n = 1$, the result follows from Lemma 1. Therefore, we will assume $n > 1$. The online algorithm finds, at time $i$, some $\tilde{y}_i \triangleq (\hat{y}_i, \tilde{y}_i^1, \ldots, \tilde{y}_i^L)$ that maximizes the following objective over valid paths $y = (y_i, \ldots, y_{i+L})$:

$$f(y) \triangleq R(y_i | \hat{y}_{[i-n,i-1]}) \quad + \quad \sum_{j=1}^{n-1} \gamma^j R(y_{i+j} | \hat{y}_{[i-n+j,i-1]}, y_{[i,i+j-1]})$$

$$+ \quad \sum_{j=n}^{L} \gamma^j R(y_{i+j} | y_{[i+j-n,i+j-1]}) \,.$$

One candidate path for the online algorithm (a) makes a transition to $y_i^*$ worth $R(y_i^* | \hat{y}_{[i-n,i-1]}) \geq 0$, (b) then follows the sequence of $n - 1$ states $y_{[i+1,i+n-1]}^*$ where transition $i + j, j \in [n - 1]$ is worth

$$\gamma^j R(y_{i+j}^* | \hat{y}_{[i-n+j,i-1]}, y_{[i,i+j-1]}^*) \geq 0 \,,$$

and (c) finally follows a sequence of $L - n + 1$ states $y_{[i+n,i+L]}^*$ where transition $i + j, j \in \{n, n+1, \ldots, L\}$ is worth $\gamma^j R(y_{i+j}^* | y_{[i+j-n,i+j-1]}^*)$. Since $\tilde{y}_i \in \mathrm{argmax}_y f(y)$ and the rewards in (a) and (b) are all non-negative, we must have

$$f(\tilde{y}_i) \geq \sum_{j=n}^{L} \gamma^j R(y_{i+j}^* | y_{[i+j-n,i+j-1]}^*) \,. \tag{5}$$

Another option available with the online algorithm is to continue following the path selected at time $i - 1$ for $L$ steps, and then make an additional arbitrary transition with a non-negative reward. Therefore, we must also have

$$f(\tilde{y}_i) \geq R(\tilde{y}_{i-1}^1 | \hat{y}_{[i-n,i-1]}) \quad + \quad \sum_{j=1}^{n-1} \gamma^j R(\tilde{y}_{i-1}^{j+1} | \hat{y}_{[i-n+j,i-1]}, \tilde{y}_{i-1}^{[j]})$$

$$+ \quad \sum_{j=n}^{L-1} \gamma^j R(\tilde{y}_{i-1}^{j+1} | \tilde{y}_{i-1}^{[j-n+1,j]}) \,. \tag{6}$$

Multiplying (5) by $1 - \gamma$ and (6) by $\gamma$, and adding the resulting inequalities, we get

$$f(\tilde{y}_i) \geq (1 - \gamma) \sum_{j=n}^{L} \gamma^j R(y_{i+j}^* | y_{[i+j-n,i+j-1]}^*) + \gamma R(\tilde{y}_{i-1}^1 | \hat{y}_{[i-n,i-1]})$$

$$+ \quad \sum_{j=2}^{n} \gamma^j R(\tilde{y}_{i-1}^j | \hat{y}_{[i-n+j-1,i-1]}, \tilde{y}_{i-1}^{[j-1]}) + \sum_{j=n+1}^{L} \gamma^j R(\tilde{y}_{i-1}^j | \tilde{y}_{i-1}^{[j-n,j-1]}) \,. \tag{7}$$

Expanding the terms of $f(\tilde{y}_i)$, we note

$$f(\tilde{y}_i) = R(\hat{y}_i | \hat{y}_{[i-n,i-1]}) + \gamma R(\tilde{y}_i^1 | \hat{y}_{[i-n+1,i]})$$

$$+ \quad \sum_{j=2}^{n} \gamma^j R(\tilde{y}_i^j | \hat{y}_{[i+j-n,i]}, \tilde{y}_i^{[j-1]}) \quad + \quad \sum_{j=n+1}^{L} \gamma^j R(\tilde{y}_i^j | \tilde{y}_i^{[j-n,j-1]}) \,. \tag{8}$$

Substituting $f(\tilde{y}_i)$ from (8) in (7), assuming zero padding as in the proof of Lemma 1, and summing over all time steps $i$, we get the inequality

$$\sum_i R(\hat{y}_i | \hat{y}_{[i-n,i-1]}) \geq \sum_i \sum_{j=n}^{L} (1 - \gamma) \gamma^j R(y_{i+j}^* | y_{[i+j-n,i+j-1]}^*) \,.$$

Defining $s = i + j$ and interchanging the two sums, we note that the right side simplifies to

$$(1 - \gamma) \sum_{j=n}^{L} \gamma^j \sum_s R(y_s^* | y_{[s-n,s-1]}^*) \,.$$

The sum of this geometric series is given by $\gamma^n - \gamma^{L+1}$, and thus setting

$$\gamma = \left( \frac{n}{L+1} \right)^{1/(L-n+1)} ,$$

we immediately conclude that the total reward obtained by the optimal offline algorithm exceeds that of the online algorithm by at most $\Theta\left( \frac{\log L}{L-n+1} \right)$ times the reward of the online algorithm, and hence we have the following bound on the competitive ratio

$$\rho \leq 1 + \Theta\left( \frac{\log L}{L-n+1} \right) .$$

$\square$

We are now ready to prove Theorem 1.

## C $n^{th}$ order chain models with diameter $\Delta$

**Theorem 1.** *The competitive ratio of Peek Search on Markov chain models of order $n$ with diameter $\Delta$ for $L \geq \Delta + n - 1$ is $\rho \leq (\gamma^{\Delta+n-1} - \gamma^{L+1})^{-1}$. Setting $\gamma = \sqrt[(L-\Delta-n+2)]{\frac{\Delta+n-1}{L+1}}$, we get*

$$\rho \leq \frac{L+1}{L-\Delta-n+2} \left( \frac{L+1}{\Delta+n-1} \right)^{(n+\Delta-1)/(L-\Delta-n+2)} = 1 + \Theta\left( \frac{\log L}{L - \tilde{\Delta} + 1} \right) .$$

*Proof.* For $\Delta = 1$, the result follows from Lemma 2. Therefore, we will assume $\Delta > 1$. As in the proof of Theorem 2, the online algorithm finds at time $i$ some $\tilde{y}_i \triangleq (\hat{y}_i, \tilde{y}_i^1, \ldots, \tilde{y}_i^L)$ that maximizes the following objective over valid paths $y = (y_i, \ldots, y_{i+L})$:

$$f(y) \triangleq R(y_i | \hat{y}_{[i-n,i-1]}) + \sum_{j=1}^{n-1} \gamma^j R(y_{i+j} | \hat{y}_{[i-n+j,i-1]}, y_{[i,i+j-1]})$$

$$+ \sum_{j=n}^{L} \gamma^j R(y_{i+j} | y_{[i+j-n,i+j-1]}) .$$

Since $\Delta > 1$, the online algorithm may not be able to jump to the desired state on the optimal offline path in one step unlike in the setting of Lemma 2, and may require $\Delta$ steps in the worst case.[5] Therefore, let $(\bar{y}_i, \ldots, \bar{y}_{i+\Delta-2})$ be an intermediate sequence of states before the online algorithm could transition to the optimal offline path and then follow the optimal algorithm for the remaining steps. Therefore, we have

$$f(\tilde{y}_i) \geq R(\bar{y}_i | \hat{y}_{[i-n,i-1]}) + \sum_{j=1}^{\Delta-2} \gamma^j R(\bar{y}_{i+j} | \hat{y}_{[i-n+j,i-1]}, \bar{y}_{[i,i+j-1]})$$

$$+ \gamma^{\Delta-1} R(y_{i+\Delta-1}^* | \bar{y}_{[i+\Delta-n-1,i+\Delta-2]})$$

$$+ \sum_{j=\Delta}^{\Delta+n-2} \gamma^j R(y_{i+j}^* | \bar{y}_{[i+j-n-1,i+\Delta-2]}, y_{[i+\Delta-1,i+j-1]}^*)$$

$$+ \sum_{j=\Delta+n-1}^{L} \gamma^j R(y_{i+j}^* | y_{[i+j-n,i+j-1]}^*)$$

$$\geq \sum_{j=\Delta+n-1}^{L} \gamma^j R(y_{i+j}^* | y_{[i+j-n,i+j-1]}^*) , \tag{9}$$

where we have leveraged the non-negativity of rewards to obtain the last inequality.

Another option available with the online algorithm is to continue following the path selected at time $i-1$ for $L$ steps, and then make an additional arbitrary transition with a non-negative reward. Therefore, we must also have

$$
\begin{aligned}
f(\tilde{y}_i) \geq R(\tilde{y}_{i-1}^1|\hat{y}_{[i-n,i-1]}) \quad &+\quad \sum_{j=1}^{n-1}\gamma^j R(\tilde{y}_{i-1}^{j+1}|\hat{y}_{[i-n+j,i-1]},\tilde{y}_{i-1}^{[j]}) \\
&+\quad \sum_{j=n}^{L-1}\gamma^j R(\tilde{y}_{i-1}^{j+1}|\tilde{y}_{i-1}^{[j-n+1,j]})\,.
\end{aligned}
\tag{10}
$$

Multiplying (9) by $1-\gamma$ and (10) by $\gamma$, and adding the resulting inequalities, we get

$$
\begin{aligned}
f(\tilde{y}_i) \quad\geq\quad & (1-\gamma)\sum_{j=\Delta+n-1}^{L}\gamma^j R(y_{i+j}^*|y_{[i+j-n,i+j-1]}^*)\;+\;\gamma R(\tilde{y}_{i-1}^1|\hat{y}_{[i-n,i-1]}) \\
+\;&\sum_{j=1}^{n-1}\gamma^{j+1}R(\tilde{y}_{i-1}^{j+1}|\hat{y}_{[i-n+j,i-1]}),\tilde{y}_{i-1}^{[j]})\;+\;\sum_{j=n}^{L-1}\gamma^{j+1}R(\tilde{y}_{i-1}^{j+1}|\tilde{y}_{i-1}^{[j-n+1,j]})\,.
\end{aligned}
$$

Expanding $f(\tilde{y}_i)$, telescoping over $i$, and defining $s=i+j$ as in Lemma 2, we get that the total reward accumulated by the online algorithm is at least $(\gamma^{n+\Delta-1}-\gamma^{L+1})$ times the total reward collected by the optimal offline algorithm since

$$
(1-\gamma)\sum_{j=\Delta+n-1}^{L}\gamma^j \;=\; \sum_{j=\Delta+n-1}^{L}(\gamma^j-\gamma^{j+1})
$$

$$
\begin{aligned}
&=\quad (\gamma^{\Delta+n-1}-\gamma^{\Delta+n})+(\gamma^{\Delta+n}-\gamma^{\Delta+n+1})+\ldots+(\gamma^{L-1}-\gamma^L)+(\gamma^L-\gamma^{L+1}) \\
&=\quad \gamma^{\Delta+n-1}-\gamma^{L+1}
\end{aligned}
$$

We immediately obtain the optimal $\gamma$ by setting the derivative with respect to $\gamma$ to 0. The optimal value turns out to be

$$
\gamma = \sqrt[(L-\Delta-n+2)]{\frac{\Delta+n-1}{L+1}}\,,
$$

which immediately yields

$$
\begin{aligned}
\rho \;\leq\; & \frac{L+1}{L-\Delta-n+2}\left(\frac{L+1}{\Delta+n-1}\right)^{(n+\Delta-1)/(L-\Delta-n+2)} \\
=\; & 1+\Theta\left(\frac{\log L}{L-\Delta-n+2}\right)\,.
\end{aligned}
$$

$\square$

Note that Theorem 1 suggests that essentially $n+\Delta-1$ steps are wasted every $L+1$ steps by the online algorithm in the sense that it may not receive any reward in these steps. However, the remaining steps fetch nearly the same reward as the optimal offline algorithm. In particular, the competitive ration $\rho$ gets arbitrarily close to 1, as $L$ is set sufficiently large compared to $\Delta+n$. That is, the performance of the online algorithm is asymptotically optimal in the peek $L$.

We now show that the result extends to the non-homogeneous setting.

## D   Non-homogeneous Markov chain models

We note that there might be multiple transitions between a pair of states during any peek window. Such transitions are considered distinct and may indeed have different rewards during the same

window. We only require that the non-discounted reward committed for every transition is "honored" at all times during the window. We have the following result.

*The competitive ratio of Peek Search on non-homogeneous (i.e. time-varying) Markov chain models of order $n$ with diameter $\Delta$ for $L \geq \Delta + n - 1$ is*

$$
\begin{aligned}
\rho &\leq \frac{L+1}{L-\Delta-n+2}\left(\frac{L+1}{\Delta+n-1}\right)^{(n+\Delta-1)/(L-\Delta-n+2)} \\
&= 1 + \Theta\left(\frac{\log L}{L-\Delta-n+2}\right),
\end{aligned}
$$

*provided the reward associated with any transition does not change for (at least) $L + 1$ steps from the time it is revealed as peek information to the online algorithm.*

*Proof.* The online algorithm maximizes the following non-stationary objective at time $i$:

$$
\begin{aligned}
f_i(y) &\triangleq R_i(y_i|\hat{y}_{[i-n,i-1]}) + \sum_{j=1}^{n-1}\gamma^j R_i(y_{i+j}|\hat{y}_{[i-n+j,i-1]}, y_{[i,i+j-1]}) \\
&\quad + \sum_{j=n}^{L}\gamma^j R_i(y_{i+j}|y_{[i+j-n,i+j-1]}),
\end{aligned}
$$

where the subscript $i$ shown with $f$ and $R$ indicates that the rewards associated with a transition may change with time $i$. Proceeding as in the proof of Theorem 1, we get

$$
\begin{aligned}
f_i(\tilde{y}_i) &\geq (1-\gamma)\sum_{j=\Delta+n-1}^{L}\gamma^j R_i(y^*_{i+j}|y^*_{[i+j-n,i+j-1]}) \\
&\quad + \gamma R_i(\tilde{y}^1_{i-1}|\hat{y}_{[i-n,i-1]}) \\
&\quad + \sum_{j=1}^{n-1}\gamma^{j+1}R_i(\tilde{y}^{j+1}_{i-1}|\hat{y}_{[i-n+j,i-1]}), \tilde{y}^{[j]}_{i-1}) \\
&\quad + \sum_{j=n}^{L-1}\gamma^{j+1}R_i(\tilde{y}^{j+1}_{i-1}|\tilde{y}^{[j-n+1,j]}_{i-1}).
\end{aligned}
$$

However, by our assumption, we can equivalently write

$$
\begin{aligned}
f_i(\tilde{y}_i) &\geq (1-\gamma)\sum_{j=\Delta+n-1}^{L}\gamma^j R_i(y^*_{i+j}|y^*_{[i+j-n,i+j-1]}) \\
&\quad + \gamma R_{i-1}(\tilde{y}^1_{i-1}|\hat{y}_{[i-n,i-1]}) \\
&\quad + \sum_{j=1}^{n-1}\gamma^{j+1}R_{i-1}(\tilde{y}^{j+1}_{i-1}|\hat{y}_{[i-n+j,i-1]}), \tilde{y}^{[j]}_{i-1}) \\
&\quad + \sum_{j=n}^{L-1}\gamma^{j+1}R_{i-1}(\tilde{y}^{j+1}_{i-1}|\tilde{y}^{[j-n+1,j]}_{i-1}).
\end{aligned}
$$

Expanding $f(\tilde{y}_i)$, summing over all $i$, and defining $s = i + j$ as in Theorem 2, we get

$$
\begin{aligned}
\sum_i R_i(\hat{y}_i|\hat{y}_{[i-n,i-1]}) &\geq \sum_i\sum_{j=\Delta+n-1}^{L}(1-\gamma)\gamma^j R_i(y^*_{i+j}|y^*_{[i+j-n,i+j-1]}) \\
&= (1-\gamma)\sum_{j=\Delta+n-1}^{L}\gamma^j\sum_s R_{s-j}(y^*_s|y^*_{[s-n,s-1]}) \\
&= (1-\gamma)\sum_{j=\Delta+n-1}^{L}\gamma^j\sum_s R_s(y^*_s|y^*_{[s-n,s-1]}),
\end{aligned}
$$

where we have again made use of the fact that reward due to any transition does not change for $L+1$ steps once revealed. The rest of the proof is identical to the analysis near the end of proof for Theorem 1. □

## E Efficient Dynamic Programs

**Theorem 2.** *Peek Search can compute a best $\gamma$-discounted path for the next $L+1$ steps, in $n^{th}$ order Markov chain models, in time $O(L|K|^n)$, where $K$ is the set of states.*

*Proof.* Let $S_i(\ell, v_{[a,b]})$ denote the set of all valid paths of length $\ell + 1$ emanating from the state $\hat{y}_{i-1}$ at time $i$, where $\ell \in \{0, 1, \ldots, L\}$, that end in the state sequence $(v_a, \ldots, v_b)$. Thus, e.g., if the directed edge $e = (\hat{y}_{i-1}, v_n)$ exists, then

$$S_i(0, v[2, n]) = \begin{cases} \{e\} & \text{if } v_{n-j} = \hat{y}_{i-j}, \ \forall j \in [n-2] \\ \emptyset & \text{otherwise} , \end{cases}$$

where $\emptyset$ is the empty set. We also denote the reward resulting from valid paths of length $\ell + 1$ that end in sequence $v_{[a,b]}$ by $\Pi_i(\ell, v[a,b])$. That is,

$$\Pi_i(\ell, v_{[a,b]}) = \max_{(y_i, \ldots, y_{i+\ell}) \in S_i(\ell, v_{[a,b]})} f_\ell(y_{[i,i+\ell]}),$$

where we define $f_\ell(y_{[i,i+\ell]})$ recursively as

$$f_\ell(y_{[i,i+\ell]}) = \begin{cases} R(y_i | \hat{y}_{[i-n,i-1]}) & \ell = 0 \\ \\ f_{\ell-1}(y_{[i,i+\ell-1]}) + \gamma^\ell R(y_{i+\ell} | \hat{y}_{[i-n+\ell,i-1]}, y_{[i,i+\ell-1]}) & \ell \in [n-1] \\ \\ f_{\ell-1}(y_{[i,i+\ell-1]}) + \gamma^\ell R(y_{i+\ell} | y_{[i-n+\ell,i+\ell-1]}) & \ell \in [n, L] \end{cases}.$$

Note that $f_L(y_{i,i+L})$ is precisely the objective optimized by Peek Search at time $i$. Now, suppose $\ell \in [n, L]$. Then, for any end sequence $v_{[2,n]}$,

$$\begin{aligned} \Pi_i(\ell, v_{[2,n]}) &= \max_{y_{[i,i+\ell]} \in S_i(\ell, v_{[2,n]})} f_\ell(y_{[i,i+\ell]}) \\ &= \max_{v_1} \max_{y_{[i,i+\ell]} \in S_i(\ell, v_{[1,n]})} f_\ell(y_{[i,i+\ell]}) , \end{aligned}$$

which may be expanded as[6]

$$\begin{aligned} & \max_{v_1 \in K} \max_{y_{[i,i+\ell]} \in S_i(\ell, v_{[1,n]})} f_{\ell-1}(y_{[i,i+\ell-1]}) + \gamma^\ell R(y_{i+\ell} | y_{[i-n+\ell,i+\ell-1]}) \\ =\ & \max_{v_1} \max_{S_i(\ell, v_{[n]})} f_{\ell-1}(y_{[i,i+\ell-1]}) + \gamma^\ell R(v_n | v_{[n-1]}) \\ =\ & \max_{v_1} \max_{S_i(\ell-1, v_{[n-1]})} f_{\ell-1}(y_{[i,i+\ell-1]}) + \gamma^\ell R(v_n | v_{[n-1]}) \\ =\ & \max_{v_1} \left( \Pi_i(\ell-1, v_{[n-1]}) + \gamma^\ell R(v_n | v_{[n-1]}) \right) . \end{aligned}$$

A similar analysis can be done for $\ell \in [n-1]$. Then, the maximizing path of length $\ell + 1$ is in the set

$$\arg \max_{v_{[2,n]} \in K} \max_{v_1 \in K} \left( \Pi_i(\ell-1, v_{[n-1]}) + \gamma^\ell R(v_n | v_{[n-1]}) \right),$$

which requires[7] checking $O(|K|^n)$ values for $v_{[n]}$. We conclude by noting that $\Pi_i$ is updated for each $\ell \in \{0, \ldots, L\}$, and thus the total complexity is $O(L|K|^n)$.

We sketch our efficient traceback procedure in Algorithm 1. In the procedure, we let $S_i^{(\ell)}, \ell \in \{0, \ldots, L\}$ be all state sequences of length $\ell + 1$ that start from state at time $i$. Thus, for instance, $S_i^{(0)}$ contains all states $y_i$ that can be reached in one step.

**Algorithm 1** Peek Search $(\gamma, L, R_i, \hat{y}_{i-n}, \ldots, \hat{y}_{i-1})$

---

**Input:** previous states $\hat{y}_{[i-n,i-2]}$ and current state $\hat{y}_{i-1}$, latency $L$, discount factor $\gamma$ and reward function $R_i(\cdot|\cdot)$

**Output:** a sequence of states that maximizes the $\gamma$-discounted reward over paths of length $(L+1)$

*Initialize rewards available in the immediate step*

---

Set $y_{i-j} = \hat{y}_{i-j}, \quad \forall j \in [n]$

$$\Pi_i(0, y_{[i-n,i-1]}, y_i) = \begin{cases} R_i(y_i|y_{[i-n,i-1]}), y_i \in S_i^{(0)} \\ 0 \qquad\qquad\qquad \text{otherwise} \end{cases}$$

*Update rewards & backpointers incrementally*

---

Define the shorthand $\quad y_{(a,b)}^{m,n} \triangleq y_{[a+m,b+n]}$

**for** $\ell = 1$ **to** $L$ for $y_{i+\ell} \in S_i^{(\ell)}$ **do**

$$\Pi_i(\ell, y_{(i,i-1)}^{\ell-n,\ell}, y_{i+\ell}) = \max_z \left( \Pi_i(\ell-1, z, y_{(i,i-1)}^{\ell-n,\ell}) + \gamma^\ell R_i(y_{i+\ell}|z, y_{(i,i-1)}^{\ell-n,\ell}) \right)$$

Store the backpointer $z_\ell^*(y_{i+\ell})$ that maximizes the score $\Pi_i(\ell, y_{(i,i-1)}^{\ell-n,\ell}, y_{i+\ell})$ above

**end for**

*Trace back a path with maximum discounted reward*

---

$\tilde{y}_{i+L} \in \underset{y_{i+L}}{\operatorname{argmax}} \max_{y_{[i+L-n,i-1+L]}} \Pi_i(L, y_{(i,i-1)}^{L-n,L}, y_{i+L+1})$

**for** $\ell = L-1$ **to** $0$ **do**

$\quad \tilde{y}_{i+\ell} = z_{\ell+1}^*(\tilde{y}_{i+\ell+1})$

**end for**

Set $\hat{y}_i = \tilde{y}_i$

---

Note that both Randomized Peek Search and Peek Reset, can compute rewards on their paths efficiently by using our procedure for Peek Search as a subroutine. For instance, Randomized Peek Search could invoke Algorithm 1 at each reset point with $\gamma$ set to 1, and follow this path until the next reset point. □

## F   Randomized Peek Search

**Theorem 3.** *Randomized Peek Search achieves, in expectation, on Markov chain models of order $n$ with diameter $\Delta$ a competitive ratio*

$$\begin{aligned} \rho &\leq 1 + \frac{\Delta + n - 1}{L + 1 - (\Delta + n - 1)} \\ &= 1 + \Theta\left(\frac{1}{L - \tilde{\Delta} + 1}\right). \end{aligned}$$

*Proof.* Recall that the randomized algorithm recomputes and follows a path that optimizes the non-discounted reward once every $L+1$ steps (which we call an *epoch*). Since the starting or reset point is chosen uniformly at random from $\{1, 2, \ldots, L+1\}$, we define a random variable $X$ that denotes the outcome of an unbiased $(L+1)$-sided dice. Let $(X = x)$ be any particular realization. Then, during epoch $i$, one option available with the online algorithm is to give up rewards in steps

$$[i * (L+1) + x, i * (L+1) + x + \Delta + n - 2]$$

to reach a state on the optimal offline path and follow it for the remainder of the epoch. Let $ON_x$ denote the total reward of the online randomized algorithm conditioned on realization $x$, and let $OPT$ be the optimal reward. Then, letting $r_t^*$ be the reward obtained by the optimal offline algorithm at time $t$ we must have

$$ON_x \quad \geq \quad OPT - \sum_i \sum_{t=i*(L+1)+x}^{i*(L+1)+x+\Delta+n-2} r_t^*. \tag{11}$$

Since $x$ is chosen uniformly at random from $[L+1]$, we also note the expected value of the second term on the right

$$
\begin{aligned}
&= \mathbb{E}_x \left( \sum_i \sum_{t=i*(L+1)+x}^{i*(L+1)+x+\Delta+n-2} r_t^* \,\Big|\, X = x \right) \\
&= \frac{1}{L+1} \sum_{x=1}^{L+1} \sum_i \sum_{t=i*(L+1)+x}^{i*(L+1)+x+\Delta+n-2} r_t^* \\
&= \frac{1}{L+1} \sum_{x=1}^{L+1} \sum_i \sum_{z=0}^{\Delta+n-2} r_{z+i*(L+1)+x}^* \\
&= \frac{1}{L+1} \sum_{z=0}^{\Delta+n-2} \left( \sum_i \sum_{x=1}^{L+1} r_{z+i*(L+1)+x}^* \right) \\
&= \frac{1}{L+1} \sum_{z=0}^{\Delta+n-2} OPT \\
&= \frac{\Delta+n-1}{L+1} OPT \ .
\end{aligned}
$$

Therefore, taking expectations on both sides of (11),

$$
\mathbb{E}_x(ON_x) \geq OPT \left( 1 - \frac{\Delta+n-1}{L+1} \right),
$$

whence the result follows immediately.

$\square$

# G Peek Reset

**Theorem 4.** *The competitive ratio of Peek Reset on Markov chain models of order $n$ with diameter $\Delta$ for latency $L$ is*

$$
\rho \quad \leq \quad 1 + \frac{2(\Delta+n)(\Delta+n-1)}{L-8(\Delta+n-1)+1} \quad = \quad 1 + \Theta\left( \frac{1}{L-8\tilde{\Delta}+1} \right) \ .
$$

*Proof.* We will assume for now that $L$ is a multiple of $4(\Delta+n-1)$. Recall that the Peek Reset algorithm works in phases with varying lengths, and takes multiple steps in each phase. Let $(i)$ denote the time at which phase $i$ begins. Then, the algorithm follows, in phase $i$, a sequence of states $\hat{y}(i) \triangleq (\hat{y}_{(i)}, \hat{y}_{(i)+1}, \ldots, \hat{y}_{T_i-1})$ that maximizes the following objective over valid paths $y = (y_{(i)}, \ldots, y_{T_i-1})$:

$$
\begin{aligned}
f(y) \quad \triangleq \quad & R(y_{(i)} | \hat{y}_{[(i)-n,(i)-1]}) \\
& + \sum_{j=1}^{n-1} R(y_{(i)+j} | \hat{y}_{[(i)-n+j,(i)-1]}, y_{[(i),(i)+j-1]}) \\
& + \sum_{j=n}^{T_i-(i)-1} R(y_{(i)+j} | y_{[(i)+j-n,(i)+j-1]}) \ ,
\end{aligned}
$$

where $T_i$ is chosen arbitrarily from the set

$$
\arg \min_{t \in [(i)+L/2+1,(i)+L]} \max_{(y_{t-n},\ldots,y_t)} R(y_t | y_{[t-n,t-1]}) \ .
$$

We define the corresponding reward

$$
x_{T_i} = \min_{t \in [(i)+L/2+1,(i)+L]} \max_{(y_{t-n},\ldots,y_t)} R(y_t | y_{[t-n,t-1]}) \ .
$$

Consider the portion of the path traced by the online algorithm from $\hat{y}_{(i)+L/2}$ to $\hat{y}_{T_i-1}$. Total number of edges on this path is $z_i = T_i - ((i) + L/2 + 1)$. We claim that the reward resulting from this sequence is at least

$$a_i = \frac{z_i - (\Delta + n - 1)}{\Delta + n} x_{T_i} .$$

This is true since, by definition of $x_{T_i}$, at each time $t \in [(i) + L/2 + 1, (i) + L]$, there is a state $y_{t-1}$ such that moving to some state $y_t$ will fetch a reward at least $x_{T_i}$. Note that a maximum of $\Delta + n - 1$ steps might have to be wasted to reach another state that fetches at least $x_{T_i}$. Thus, a reward of $x_{T_i}$ is guaranteed every $\Delta + n$ steps. While there are $z_i$ steps in this sequence, at most $\Delta + n - 1$ steps may be left over as residual edges that do not fetch any reward if $z_i$ is not a multiple of $\Delta + n$. Since the online algorithm optimized for total non-discounted reward, it must have considered this alternative subsequence of steps for the interval pertaining to $z_i$.

Next consider the portion traversed by the online algorithm from $\hat{y}_{T_i}$ to $\hat{y}_{(i)+L}$ in the next phase $(i+1)$. This phase starts at time $T_i$. By an argument analogous to previous paragraph, the online algorithm collects from this sequence an aggregate no less than

$$b_i = \frac{(i) + L - T_i - (\Delta + n - 1)}{\Delta + n} x_{T_i} .$$

Thus, the reward accumulated by the online algorithm in these two portions is at least

$$a_i + b_i = \frac{L - 4(\Delta + n - 1)}{2(\Delta + n)} x_{T_i} .$$

Summing over all phases, we note that the total reward gathered by the online algorithm is

$$\sum_i f(\hat{y}(i)) \geq \frac{L - 4(\Delta + n - 1)}{2(\Delta + n)} \sum_i x_{T_i} . \tag{12}$$

Let $f(y^*(i))$ be the reward collected by the optimal offline algorithm in phase $i$. Since the online algorithm optimizes for the total reward, one possibility it considers is to forgo reward in the first $(\Delta + n - 1)$ steps in each phase in order to traverse the same sequence as the optimal algorithm in the remaining steps. Thus, we must have

$$\sum_i f(\hat{y}(i)) \geq \sum_i f(y^*(i)) - (\Delta + n - 1) \sum_i x_{T_i} . \tag{13}$$

Combining (12) and (13), we note for even $L$

$$\frac{\sum_i f(y^*(i))}{\sum_i f(\hat{y}(i))} \leq 1 + \frac{2(\Delta + n)(\Delta + n - 1)}{L - 4(\Delta + n - 1)} .$$

Accounting for $L$ that are not multiples of $4(\Delta + n - 1)$, we conclude the competitive ratio of Peek Reset is

$$\rho \leq 1 + \frac{2(\Delta + n)(\Delta + n - 1)}{L - 8(\Delta + n - 1) + 1} .$$

$\square$

# H   Lower Bounds

**Theorem 5.** *The competitive ratio of any deterministic online algorithm on $n^{th}$ order (time-varying) Markov chain models with diameter $\Delta$ for latency $L$ is greater than*

$$1 + \frac{\tilde{\Delta}}{L} \left( 1 + \frac{\tilde{\Delta} + L - 1}{(\tilde{\Delta} + L - 1)^2 + \tilde{\Delta}} \right) .$$

*In particular, when $n = 1$, $\Delta = 1$, the ratio is larger than*

$$1 + \frac{1}{L} + \frac{1}{L^2 + 1} .$$

*Proof.* We motivate the main ideas of the proof for the specific setting of $n = 2$ and unit diameter. The extension to general $n$ and unit diameter is then straightforward. Finally, we conjure an example to prove the lower bound for arbitrary $n$ and $\Delta$ via a prismatic polytope construction.

First consider the case $n = 2$ and $\Delta = 1$. We design a $3 \times (L + 3)$ matrix initialized as shown below: each row corresponds to a different state, each column corresponds to a time, "?" indicates that the corresponding entry is not known since it lies outside the current peek window of length $L + 1$, and $a > 0$ is a variable that will be optimized later.

$$\begin{bmatrix} \square 0 & 1 & a & a & \dots & a & ? & ? \\ 0 & 1 & a & a & \dots & a & ? & ? \\ 0 & 1 & \underbrace{a & a & \dots & a}_{(L-1)\ \text{terms}} & ? & ? \end{bmatrix} \tag{14}$$

The box in front of the first entry indicates that the online algorithm made a transition to state 1 from a dummy start state " $*$ " and is ready to make a decision in the current step $t = 0$ about whether to continue staying in state 1, or transition to either state 2 or 3. At time $t = 0$, the rewards for the next $L + 1$ steps are identical, so without loss of generality, let the online algorithm choose the first state, get 0 as reward, and move to the next time $t = 1$. An additional column is revealed and we get the following snapshot.

$$\begin{bmatrix} \square 0 & \square 1 & a & a & \dots & a & 0 & ? \\ 0 & 1 & a & a & \dots & a & 2a & ? \\ 0 & 1 & a & a & \dots & a & 2a & ? \end{bmatrix} \tag{15}$$

Since $n = 2$, we may enforce the following second order Markov dependencies for $t \geq 1$: any state $s \in \{1, 2, 3\}$ yields zero reward unless the previous two states $s', s'' \in \{1, 2, 3, *\}$ were such that $s' \in \{*, s\}$ and $s'' = s$. If this condition is true, then the algorithm receives the current entry pertaining to $s$ as the reward. In other words, other than the special case of dummy start state being one of the states, the algorithm receives the reward only if $s$ is same as the previous two states.

Suppose the online algorithm selects state 1 again at $t = 1$. Then it collects reward 1, and another column is revealed as shown below.

$$\begin{bmatrix} \square 0 & \square 1 & \square a & a & \dots & a & 0 & 0 \\ 0 & 1 & a & a & \dots & a & 2a & 0 \\ 0 & 1 & a & a & \dots & a & 2a & 0 \end{bmatrix} \tag{16}$$

In this scenario, the maximum reward the online algorithm can fetch, during its entire execution, is at most $1 + (L - 1)a$. To see this note that this is exactly the reward the algorithm gets if it sticks to state 1 at all subsequent times $t$. If, however, it were to jump to any other state and continue with it for at least one step, then it would lose rewards in successive steps due to second order dependency, for a total loss of reward $2a$. All other possibilities incur a loss greater than $2a$. This loss offsets the additional $2a$ reward available with states other than 1. On the other hand, the offline algorithm would select a state $s \in \{2, 3\}$ from the very beginning, and thus receive $1 + (L + 1)a$ in total. The competitive ratio in this scenario, therefore, turns out to be

$$r_1 = \frac{1 + (L + 1)a}{1 + (L - 1)a} = 1 + \frac{2a}{1 + (L - 1)a} \ .$$

Suppose instead the online algorithm transitions to some state $s \in \{2, 3\}$ at $t = 1$. We assume without loss of generality that the algorithm transitions to state 2. The last column is then revealed as follows.[8]

$$\begin{bmatrix} \Box 0 & \Box 1 & a & a & \dots & a & 0 & 0 \\ 0 & 1 & \Box a & a & \dots & a & 2a & 0 \\ 0 & 1 & a & a & \dots & a & 2a & a \end{bmatrix} \tag{17}$$

Note that the online algorithm loses on rewards 1 and $a$ in successive steps due to transition. The maximum total reward possible in this case is $La$ regardless of whether the online algorithm makes a transition to other states, or sticks with state 2 subsequently. The offline algorithm, in contrast, would receive all rewards available in state 3. Thus, the ratio in this scenario is

$$r_2 = \frac{1 + (L+2)a}{La} = 1 + \frac{1 + 2a}{La} \; .$$

Combining the two cases, the competitive ratio of the online algorithm is at least $\min\{r_1, r_2\}$, and thus we could set $r_1 = r_2$ and solve for $a$.

We can extend this analysis to the general $n \geq 1$ setting with unit diameter easily. We design a $3 \times (L+3)$ matrix with the same row initialization as in (14). Also, we assume that prior to time $t = 0$, only zero reward transitions were available between some dummy states[9] for both the online and the offline algorithms. We denote the set of these dummy states by $**$. We enforce the following $n^{th}$ order Markov dependencies for $t \geq 1$: any state $s \in [m]$ yields zero reward unless the previous $n$ states were same as $s$ or had a prefix consisting only of states in $**$ followed by $s$ in the remaining time steps. If this condition is satisfied, the algorithm receives the current entry pertaining to $s$ as reward.

The evolution of the reward matrix is as follows. Assuming state 1 was selected at $t = 0$, we let column $L + 2$ have all entries in rows 2 and 3 to $na$ (instead of $2a$ that we set in (15)) at $t = 1$. Finally, if the online algorithm selects state 1 at $t = 1$, we set the last column to all zeros at time $t = 2$ as in (16); otherwise, we set first two entries in the last column to 0, and $a$ in the last row as in (17).

Reasoning along the same lines as before, the competitive ratio of the online algorithm is at least

$$\min \left\{ 1 + \frac{na}{1 + (L-1)a}, 1 + \frac{1 + na}{La} \right\} \; . \tag{18}$$

We set

$$1 + \frac{na}{1 + (L-1)a} = 1 + \frac{1 + na}{La} \; ,$$

whereby

$$a = \frac{n + L - 1 + \sqrt{(n + L - 1)^2 + 4n}}{2n} \; .$$

Substituting this value for $a$ in (18), and leveraging that

$$a < \frac{n + L - 1}{n} + \frac{1}{n + L - 1} \; ,$$

we note the competitive ratio is at least

$$1 + \frac{n}{L} \left( 1 + \frac{n + L - 1}{(n + L - 1)^2 + n} \right) \; . \tag{19}$$

The foregoing analysis may be visualized geometrically in terms of a triangle, with each vertex corresponding to a state. The rewards for initial $L + 1$ steps are all same, and thus the online algorithm does not have preference for any state initially. Without loss of generality, as soon as it selects state 1 (with all rewards at time $t = 0$ being 0), the rewards for time step $L + 2$ are chosen at $t = 1$ such that states 2 and 3 would fetch reward $na$ while state 1 will fetch none. The online algorithm could either stay with state 1 and get a suboptimal total reward or jump to an adjacent vertex or state, which would not yield reward for $n$ steps.

We now extend this analysis to accommodate any finite $\Delta \geq 1$. Toward that goal, we consider a $\Delta$-dimensional prismatic polytope[10] with a triangular base (i.e. having 3 vertices). Each vertex of the polytope corresponds to a state, and the maximum distance between two vertices is exactly $\Delta$. Moreover, for every vertex there is some vertex at distance $d$ for each $d \in [\Delta]$. The polytope is completely symmetric with respect to all the vertices, and we again set rewards for the first $L + 1$ steps at all vertices to be the same as before.

Without loss of generality, we again assume that the online algorithm starts at some state 1 (arbitrary labeled). At the next time step, the reward at all vertices that are at a distance $d$ from this vertex is set to $(n + d - 1)a$. Thus, the vertices adjacent to state 1 have reward $na$ in column $(L + 2)$ since they lie at distance $d = 1$, while the reward for state 1 in this column is 0. Thus, the maximum reward is available at distance $d = \Delta$ from state 1, however, the online algorithm will need to make $\Delta$ steps to reach such a state, and then wait another $n - 1$ steps before availing this reward. Thus, effectively, $\tilde{\Delta} = n + \Delta - 1$ steps are wasted that the offline algorithm could fully exploit due to prescience. Proceeding along same lines as before, and replacing $n$ with $\tilde{\Delta}$ in (19), we conclude that the competitive ratio of any deterministic online algorithm on our construction is at least

$$1 + \frac{\tilde{\Delta}}{L}\left(1 + \frac{\tilde{\Delta} + L - 1}{(\tilde{\Delta} + L - 1)^2 + \tilde{\Delta}}\right) .$$

$\square$

**Theorem 6.** *For any $\epsilon > 0$, the competitive ratio of any randomized online algorithm, that is allowed latency L, on $n^{th}$ order (time-varying) Markov chain models with $\Delta = 1$ is at least*

$$1 + \frac{(1 - \epsilon)n}{L + \epsilon n} .$$

*For a general diameter $\Delta$, the competitive ratio is at least*

$$1 + \frac{\left(2^{\Delta - 1}\lceil 1/\epsilon \rceil - 1\right)n}{2^{\Delta - 1}\lceil 1/\epsilon \rceil L + n} .$$

*Proof.* First consider the unit diameter setting (i.e. $\Delta = 1$). We design a matrix with $\lceil 1/\epsilon \rceil$ rows and $L + 2$ columns. The first column consists of all zeros, the next $L$ columns contain all ones, and the last column contains all zeros except one randomly chosen row that contains $n$. We again enforce the Markov dependency structure described in the proof of Theorem 5 for all states (or rows) in $[\lceil 1/\epsilon \rceil]$.

The optimal offline algorithm knows beforehand which row $q$ contains $n$ in the last column, and thus collects a total reward $L + n$. On the other hand, any randomized online algorithm chooses this row at $t = 0$ with only probability $\epsilon$. Selecting any other row at $t = 0$ may fetch a maximum reward of $L$ accounting for all the possibilities including sticking to this row subsequently, or moving to $q$ in one or more transitions. Since the randomized algorithm is assigned at time $t = 0$ with the remaining probability $(1 - \epsilon)$ to some row other than $q$, its expected reward cannot exceed

$$\epsilon * (L + n) + (1 - \epsilon) * L = L + \epsilon n.$$

Thus, when $\Delta = 1$, the competitive ratio for any randomized online algorithm is at least $\dfrac{L + n}{L + \epsilon n}$ as claimed. For the general setting, we consider a $\Delta$-dimensional prismatic polytope with the base containing $\lceil 1/\epsilon \rceil$ vertices. In addition to the usual prismatic polytope topology (assuming bidirectional edges between any pair of adjacent vertices), we add edges so that vertices on each face are strongly connected, i.e., directed edges in both directions connect all pairs of vertices that lie on a face. The polytope contains $u = 2^{\Delta - 1}\lceil 1/\epsilon \rceil$ states in total. We design a matrix having these many rows and $L + 2$ columns as before. Any randomized online algorithm has only a $1/u$ probability of getting the maximum possible $L + n$ reward (due to selecting $q$ and sticking with it), and must forfeit a reward no less than $n$ with the remaining probability. Thus, the expected reward cannot exceed

$$(L + n)/u + (1 - 1/u) * L = L + n/u,$$

while the maximum possible reward is $L + n$. Thus the competitive ratio is at least $\dfrac{L + n}{L + n/u}$ which simplifies to the result stated in the problem statement. $\square$

## Footnotes

[3]Note that for first order fully connected models, Peek Search recovers an algorithm introduced by [34] in the context of their single server online allocation setting. Similarly, Peek Reset generalizes the Intermittent Reset algorithm due to [34], while Randomized Peek Search generalizes a randomized algorithm from [34] to higher order settings for $\Delta \geq 1$. Our algorithms hinge on a novel adaptive optimization perspective. We therefore emphasize the role of optimization in our analysis.

[4]One way to accomplish this is by adding a sequence of $L + 1$ dummy tokens, that fetch only zero rewards, at the beginning and another sequence at the end of the input to be decoded. Alternatively, we can introduce a dummy start state that transitions to itself $L$ times with zero reward and produces a fake output in each transition, and then makes a zero reward transition into the true start state, whence actual decoding happens for $T$ steps followed by repeated transitions into a dummy end state that fetches a zero reward).

[5]The online algorithm may require less than $\Delta$ steps depending on its current state, however, we perform a worst case analysis and therefore, our result holds even if fewer than $\Delta$ steps may suffice to reach the optimal path at some point during the execution of the online algorithm.

[6]We simply write $S_i(\ell, v)$ instead of $y_{[i,i+\ell]} \in S_i(\ell, v)$ in order to improve readability at the expense of abuse of notation.

[7]In addition to *backpointer* information that is required to determine a maximizing path as in the Viterbi algorithm once the construction of table for bookkeeping $\Pi_i$ is completed. Construction of table requires $O(L|K|^n)$ time which dominates the $O(L)$ time required for computing the path from the backpointers.

[8] Note that since our objective here is to prove a lower bound, we would like the competitive ratio to be as high as possible. It might be tempting to set a reward larger than $a$ for state 3 in the last column. That would imply both the online and the offline algorithms could receive an additional reward worth $a$. This, however, would not improve the competitive ratio for the simple reason that for positive x, y, and c, $\frac{x + c}{y + c} > \frac{x}{y}$ only if $x < y$ (we instead have $x > y$ since $r_2 > 1$).

[9]Another way to enforce the same effect, without the dummy states, is to add additional $n$ columns, with all zero rewards for all the actual states, prior to time $t = 0$.

[10]Note that a $d$-dimensional prismatic polytope is constructed from two $(d$ - $1)$-dimensional polytopes, translated into the next dimension.