[Reviews · NeurIPS 2019]

Reviewer 1



Originality: As far as I know, this is the first paper to study the theory of online Markov decoding rigorously. Quality: The results our sound. Clarity: The paper is well-written overall. However the lower bounds are in a form that makes it a bit hard to compare them with the upper bounds. Significance: Markov decoding is an important scientific tool. Designing fast algorithms for this problem is imperative in my humble opinion. --- Edit --- Lowered score following comments from other reviewers.

Reviewer 2



The algorithm uses a discounted reward for lookahead observed tokens (of size L) and runs dynamic programming (similar to Viterbi) to decode the current state. The discounted reward is the key to obtain the provable bounds, however, I believe the authors have not emphasized its importance enough and it is more implicit inside the proofs. At line 228, authors mentioned beam search, similar to Viterbi, requires to compute the optimal path of the full sequence. Beam search is a greedy approach and for beam size of k, only looks at the k top recent paths for decoding. In general, I liked the idea, but I guess the paper requires polishing and better presentation to deliver its message more clearly. The paper could benefit from having a clear algorithm (pseudo-code). ======== I increased my score based on the provided feedback and other reviews.

Reviewer 3



The authors propose three online inference methods, i.e., peek search, randomized peek search, and peek reset, for Markov chain models. They also use a proof framework to prove the competitive ratio for each method. The proof method first computes the lower bound of ON, so that gets the relationship between the ON and OPT. And then it uses the properties of geometric series to find an upper bound of competitive ratio. The authors also design a dynamic programming method to efficiently compute reward accumulation, and prove its complexity. Finally, they prove the lower bounds of competitive ratio for general deterministic online algorithms. In Section 7, they compare their method, i.e., peek search, with another online inference method OSA on a genome dataset. I have to admit that I am not familiar with this area, so can only go through a part of the proof, and I am not able to evaluate the originality and quality of this work. Pros: 1. The paper is well written and easy to follow. The proof is clear and thorough. 2. I think the proof framework should be useful for proving similar problems. 4. The proposed methods have solid theoretical guarantees. 3. I believe that the lower bounds are important, which can give guidance to design new online inference method. Cons: 1. The paper mentions adversarial rewards a few times, but I did not see further discussion or proof of this topic. 2. It would be better if the authors can provide more details about how to compute the sum of geometric series, and how to set the \gamma in the appendix. 3. In the experiments, the authors only use one dataset, and show the results of peek search. It would be better if the authors can show results on more dataset and results of randomized peek search and peek reset. =============================== I have read the authors' responses, and other reviewers' comments. I did not change my score. I agree with the other reviewers that the theorems in this paper are interesting. The main reason that I did not change the score is that I am not quite familiar with this area, so I cannot evaluate the significance of this paper. I do not want to overestimate its value.

[Author Response · NeurIPS 2019]



Genome sequence results (all algorithms)

| | log-probability | tag accuracy (%) |
|---|---|---|
| **Viterbi** | -117.29 +/- .53 | 97.4 +/- .02 |
| **PS (L=1)** | -117.40 +/- .54 | 97.0 +/- .01 |
| **PS-3 (L=1)** | -117.40 +/- .54 | 97.0 +/- .02 |
| **PS (L=2)** | -117.34 +/- .54 | 97.2 +/- .01 |
| **PS-3 (L=2)** | -117.34 +/- .54 | 97.2 +/- .01 |
| **PS (L=3)** | -117.33 +/- .54 | 97.3 +/- .02 |
| **PS-3 (L=3)** | -117.33 +/- .54 | 97.3 +/- .02 |

Table 1: Part-of-speech tagging on Brown data ($PS-3 \triangleq PS$ approximated with beams (size 3))

We are grateful to the reviewers for their feedback. We address all their comments below.

**Reviewer #1:** Thank you very much for a thoughtful review, for acknowledging our theoretical and algorithmic contributions, and for emphasizing the practical significance of our work. We will fix the typo on line 142.

**Reviewer #2:** Thank you very much for your feedback. We provide additional results to address all your concerns.
(1) **Results with Peek Reset (PR) and Randomized Peek Search (RPS)**. Please see the figure above for performance of these algorithms. Since PR, by design, makes sense for $L$ large enough (exceeding 5), so we show its results for $L \geq 5$. Clearly, both the methods perform much better than OSA. We observe that Peek Search (PS) performs better than PR and RPS for $L < 20$. PR performed very well for larger $L$ (not shown in figure) as expected. In particular, the scaled log-probabilities under PR for $L = 50$ and $L = 100$ were observed, respectively, to be -39.69 and -39.56. Moreover, the decoded sequences agreed with Viterbi on 97.32% and 98.68% of the sites respectively.
(2) **Results with Beam Search of size $k$ (BS-$k$)**. Please note that despite efficient greedy path expansion, BS-$k$ has high latency (i.e., 73384 on genome sequence) since label is not known for any observation until the $k$-greedy paths are computed for entire sequence and backpointers are traced back to the start. We found that BS-2 performed worse than PS for $L \geq 5$. Also, BS-3 recorded log prob.-39.61 and decoding agreement 97.73% (worse than PR with $L = 100$). Based on your comment about BS-$k$, we also explored approximating discounted paths in PS by few beams, thereby combining strengths of both PS and BS-$k$. This algorithm performed well on standard Brown data (Table 1).
(3) **Results with Peek Search when $\gamma = 1$**. PS with $\gamma = 1$ obtained a scaled log-probability of -45.86 (and 58.8% decoding match with Viterbi), however the performance was similar to PS for larger $L$. This is expected since $\gamma$ approaches 1 as $L$ increases. Please note that PS is the only algorithm with $\gamma < 1$, and both RPS and PR do not perform any discounting (i.e. $\gamma = 1$). Based on your review, we will expand on the role of $\gamma$ for PS and include these results.
(4) **Results on short sequences.** We found PS to work exceptionally well on short Markov chains as well (both in terms of log probability and test accuracy). In particular, on the Brown data (average sentence length only 20.25), we achieved near-optimal performance under very low latency (Table 1).
(5) **Pseudocode.** Please see Appendix E for pseudocode of PS. We will add pseudocode for RPS and PR as well.

**Reviewer #3:** Thank you very much for providing a nice summary of our contributions, and for your suggestions.
(1) **Summing the geometric series, and setting $\gamma$ in the Proof of Theorem 1**. We note that $(1 - \gamma) \sum_{j=\Delta+n-1}^{L} \gamma^j$

$$= \sum_{j=\Delta+n-1}^{L} (\gamma^j - \gamma^{j+1}) = (\gamma^{\Delta+n-1} - \gamma^{\Delta+n}) + (\gamma^{\Delta+n} - \gamma^{\Delta+n+1}) + \ldots + (\gamma^{L-1} - \gamma^L) + (\gamma^L - \gamma^{L+1})$$

$$= \gamma^{\Delta+n-1} - \left( (\gamma^{\Delta+n} - \gamma^{\Delta+n}) + \ldots + (\gamma^L - \gamma^L) \right) - \gamma^{L+1} = \gamma^{\Delta+n-1} - \gamma^{L+1} \triangleq h(\gamma)$$

We immediately obtain the optimal $\gamma$ by setting the derivative, i.e. $h'(\gamma) = (\Delta + n - 1)\gamma^{\Delta+n-2} - (L + 1)\gamma^L$, to 0.
(2) **Adversarial rewards**. Our algorithms accommodate adversarial settings, since our proofs do not require any distributional assumptions on the rewards for any $(L + 1)$-long peek window: they may be chosen in an arbitrary (e.g. non-stochastic, possibly adversarial) manner, including when adapting to time (Appendix D).
(3) **Performance of Peek Reset and Randomized Peek Search**. Please see the figure for these results.
(4) **More experiments (another dataset)**. We also include results of our experiments on the standard Brown data (task is to tag sentences with their parts-of-speech). Since explicit train-test sets are not provided, we formed 5 random partitions each having 80% train and 20% test sentences. We trained a separate HMM model for each partition. We report the average test accuracy and standard deviation results (Table 1). Even with very small $L$, Peek Search nearly matched the performance of Viterbi. Moreover, similar results were obtained when we used 3 beams to greedily expand discounted paths in each $(L + 1)$-long peek window. Thus, we can combine strengths of Beam Search with Peek Search.

[Meta-Review · NeurIPS 2019]

After discussion and the author response, the reviewers landed on a consensus that the paper would be a good addition to the NeurIPS program. The author feedback provides a more thorough analysis of the experiments, which should be included in the final version.